# Bistatic Radar Scattering from Non-Gaussian Height Distributed Rough Surfaces

**Ying Yang [1], Kun-Shan Chen [2],\*** and **Suyun Wang [3]**

1   School of Electronic and Optical Engineering, Nanjing University of Science and Technology, Nanjing 210094, China
2   College of Geomatics and Geoinformation, Guilin University of Technology, Guilin 541004, China
3   National Institute of Information and Communications Technology (NICT), Tokyo 184-0015, Japan
*   Correspondence: chenks@glut.edu.cn

**Abstract:** In modeling a rough surface, it is common to assume a Gaussian height distribution. This hypothesis cannot describe an eventual asymmetry between crests and troughs of natural surfaces. We analyzed the bistatic scattering from a rough surface with non-Gaussian height distributions using the Kirchhoff scattering theory. Two extreme cases of Gamma-distributed surfaces were compared in particular: exponential and Gaussian distributions. The bistatic angular dependence was examined under various root mean square (*RMS*) heights and power spectrum densities. Contribution sources to the coherent and incoherent scattering components were singled out relating to the surface height distribution. For an exponential height surface, the coherent scattering strengthens even as the surface becomes rough. The non-Gaussian effect on the incoherent scattering is connected with surface power spectrum density. The height distribution impacts differ in the different regions of the bistatic scattering plane and thus complicate the differentiation of the scattering patterns due to height distribution.

**Keywords:** height probability density; non-Gaussian; rough surface; bistatic scattering; power spectrum density

## 1. Introduction

Understanding the electromagnetic scattering from a rough surface is vital for microwave sensing of land and ocean. In the past several decades, considerable efforts have been made to elucidate the scattering processes through theoretical modeling and numerical simulations. In modeling a rough surface, it is common to assume a Gaussian height distribution that suggests each point on the rough surface is irrelevant in the height direction, which is often far from realistic. A natural rough surface is frequently a non-Gaussian process because of natural forces. Two rough surfaces can have the same correlation function but different height distributions, or vice versa. While the non-Gaussian effect on the scattering characteristics has been well recognized, most previous works studying the radar scattering of a rough surface assumed a Gaussian height. We considered the height probability density function (HPD) non-Gaussian, to which a much smaller body of study is dedicated.

Thus far, limited studies have focused on Gaussian and non-Gaussian rough surface characteristics. For instance, Newland used the fast fourier transformation(FFT) method and prescriptive power spectral density (PSD) to generate a Gaussian HPD surface [1]. Franceschetti applied the Kirchhoff-fractal electromagnetic model to examine the surface's statistics properties with Gaussian HPD [2]. The work in [3] presented the statistical modeling of radar scattering from the ocean surface, whose slope and height assumed a Gaussian process. The research in [4] reported, in the optical region, the dependence of speckle contrast on the surface roughness where the surface height fluctuations were Gaussian. The assumption of Gaussian HPD is accepted to some extent because it is easy

to provide a simple analytical solution. However, in practice, many natural or engineered surfaces are frequently non-Gaussian processes. Therefore, the existence of a non-Gaussian effect on scattering characteristics should be taken into consideration.

Recent studies revealed that non-Gaussian HPD surfaces strongly influence scattered signal and radar imaging statistics [5]. However, to our knowledge, minimal study has been focused on the bistatic scattering for non-Gaussian HPD surfaces. Ocean and sea-ice surfaces might be two distinctive non-Gaussian surfaces [6,7]. As proposed in [7], sea ice presented negative exponential height distribution. These things considered, non-Gaussian rough surfaces are common in machining processes such as honing, grinding, and milling [8,9]. It was reported in [10,11] that material surfaces might change from a Gaussian HPD to a non-Gaussian HPD when the material surfaces are subjected to processes such as wear and friction. Furthermore, some material surfaces are designed with non-Gaussian height distribution, including piston surfaces [12], asphalt roads [13], and water channels [14].

It was shown [15] that surface height distribution is the dominant factor in determining the scattering coefficient's coherent component. The incoherent scattering also depends on the height distribution and the roughness relative to the radar wavelength. The coherent Strehl factor, defined as the ratio of the coherent intensity with and without surface roughness, depends on the surface height probability density. For a smooth surface, the coherent Strehl factor is independent of HPD, but it highly depends on the HPD when the surface roughness is moderate to large [16]. Beckmann presented the scattering behaviors for non-Gaussian surfaces by the first-order Kirchhoff model [17]. The authors of [18] reported wave scattering on rough surfaces with alpha-stable non-Gaussian height distribution under the first-order Kirchhoff and small-slope approximations. A simulation of wave scattering from a one-dimensional non-Gaussian HPD surface [19] found that the coherent scattering is higher for a non-Gaussian than a Gaussian HPD surface for both HH and VV polarizations. The quantitative characterization of non-Gaussian rough surfaces is described in [20].

The rest of the paper is organized as follows. Section 2 briefly provides two key statistics that describe a randomly rough surface: surface height distribution and power spectrum density. We considered both Gaussian and exponential distributions for surface height distribution, representing two extremes of Gamma distributions. As for power spectrum density, we considered Gaussian and exponential PSDs. Section 3 formulates the scattering problem using the Kirchhoff scattering theory, which involves univariate and bivariate height distributions. Multiple scattering was not considered in this study but will be included in a future study. Section 4 presents the surface roughness and angular dependences of the bistatic scattering in line with a non-Gaussian effect. Finally, Section 5 draws a summary of this study.

## 2. The Statistical Description of Rough Surfaces

Considering the rough surface as stationary and ergodic, the height distribution and the power spectral density are perhaps two of the most important statistical descriptors in wave scattering.

### 2.1. Height Distribution

For Gaussian height distribution, it is well known that the univariate and bivariate distributions are of the forms:

$$p_g(\zeta) = \frac{1}{\sqrt{2\pi}\sigma} e^{-\zeta^2/2\sigma^2} \tag{1}$$

$$p_g(\zeta, \zeta') = \frac{1}{2\pi\sigma^2\sqrt{1-\rho^2}} \exp\left\{-\frac{\zeta^2 - 2\rho\zeta\zeta' + \zeta'^2}{2\sigma^2(1-\rho^2)}\right\} \tag{2}$$

where $\sigma$ is the RMS height and $\rho$ is the autocorrelation function.

The univariate and bivariate exponential distributions are given by

$$p_e(\zeta) = \frac{1}{\sigma} \exp\left[-\frac{\zeta}{\sigma}\right], \zeta \geq 0 \tag{3}$$

$$p_e(\zeta, \zeta') = \frac{1}{\sigma^2(1-\rho)} \exp\left[-\frac{\zeta + \zeta'}{\sigma(1-\rho)}\right] I_0\left(\frac{2\sqrt{\rho\zeta\zeta'}}{\sigma(1-\rho)}\right) \tag{4}$$

where $\sigma$ is the RMS height and $\rho$ is the autocorrelation function; $I_0$ is the zeroth-order modified Bessel function.

### 2.2. Power Spectral Density

In this study, we considered the Gaussian and exponential PSDs for their common use in modeling the wave scattering of rough surfaces. The Gaussian and exponential PSDs indeed represent two extremes of the roughness distribution over spatial wavenumber in terms of bandwidth [21].

For a Gaussian-correlated surface, the autocorrelation function and PSD are

$$\rho_g(r) = \exp\left(-\frac{r^2}{l^2}\right) \tag{5}$$

$$S_g(\mathbf{K}) = \frac{\ell^2}{2} \exp\left(-\frac{\mathbf{K}^2\ell^2}{4}\right) \tag{6}$$

For an exponential-correlated surface, the correlation function and PSD are

$$\rho_e(r) = \exp\left(-\frac{|r|}{l}\right) \tag{7}$$

$$S_e(\mathbf{K}) = \frac{\ell^2}{\left(1 + \mathbf{K}^2\ell^2\right)^{3/2}} \tag{8}$$

where $\mathbf{K} = (K_x, K_y)$ is a spatial wavenumber vector with $K_x, K_y$ representing the wavenumber components in $x$ and $y$ directions with $K = \sqrt{K_x^2 + K_y^2}$; $\ell$ is the correlation length, and both variables are represented in the unit of radar wavelength.

## 3. Formulation of the Scattering Problem

### 3.1. Scattered Field

Referring to Figure 1, assume a plane wave impinges onto a rough, dielectric surface which scatters waves up into the upper medium and down into the lower medium in the incident plane, with the electric and magnetic fields written as

$$\vec{E}^i = \hat{p} E_0 \exp\left[-j\left(\vec{k}_i \cdot \vec{r}\right)\right] \tag{9}$$

$$\vec{H}^i = \frac{1}{\eta}\hat{k}_i \times \vec{E}^i \tag{10}$$

where $j = \sqrt{-1}$; $\hat{p}$ is the unit polarization vector; $E_0$ is the amplitude of the incident electric field; and $\eta$ is the intrinsic impedance of the upper medium.

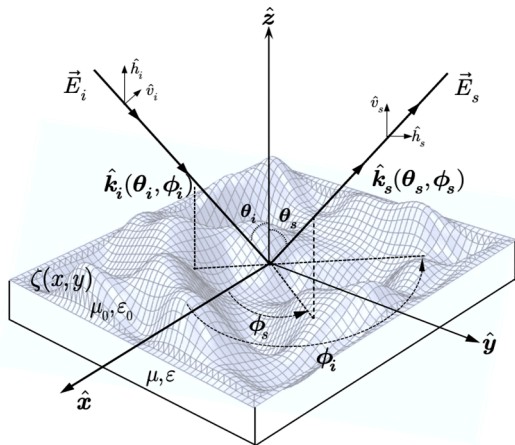

**Figure 1.** Geometry of wave scattering from a rough surface $\zeta(x,y)$.

The wave number vectors in incident and scattering directions are, respectively, defined as follows:

$$\vec{k}_i = k\hat{k}_i = \hat{x}k_{ix} + \hat{y}k_{iy} + \hat{z}k_{iz};$$
$$k_{ix} = k\sin\theta_i\cos\phi_i, k_{iy} = k\sin\theta_i\sin\phi_i, k_{iz} = -k\cos\theta_i \quad (11)$$

$$\vec{k}_s = k\hat{k}_s = \hat{x}k_{sx} + \hat{y}k_{sy} + \hat{z}k_{sz};$$
$$k_{sx} = k\sin\theta_s\cos\phi_s, k_{sy} = k\sin\theta_s\sin\phi_s, k_{sz} = k\cos\theta_s \quad (12)$$

where $k = 2\pi/\lambda$ is wavenumber, and $\lambda$ is wavelength.

Under the Kirchhoff approximation, the estimation of surface tangential fields is in order. We may define a local coordinate system $(\hat{t}, \hat{d}, \hat{k}_i)$ shown in Figure 2, with

$$\hat{t} = \frac{\hat{k}_i \times \hat{n}}{\left|\hat{k}_i \times \hat{n}\right|}, \; \hat{d} = \hat{k}_i \times \hat{t}, \; \hat{k}_i = \hat{t} \times \hat{d} \quad (13)$$

where the surface normal vector is given by

$$\hat{n} = -\frac{\nabla\zeta}{|\nabla\zeta|} = \frac{-\hat{x}\zeta_x - \hat{y}\zeta_y + \hat{z}}{\sqrt{1 + \zeta_x^2 + \zeta_y^2}} \quad (14)$$

with $\zeta_x = \partial\zeta/\partial x, \zeta_y = \partial\zeta/\partial y$ being the surface slopes along x and y directions, respectively, and are estimated by the stationary phase approximation:

$$\zeta_x = -\frac{k_{sx} - k_x}{k_{sz} - k_z}, \zeta_y = -\frac{k_{sy} - k_y}{k_{sz} - k_z} \quad (15)$$

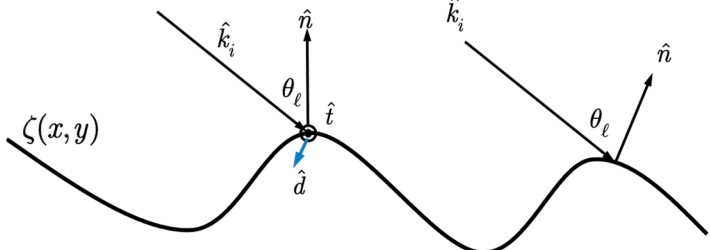

**Figure 2.** A local orthonormal coordinate $(\hat{t}, \hat{d}, \hat{k}_i)$.

Once the surface fields are obtained, the far-zone scattered field is calculated by the Stratton–Chu formula:

$$E_{qp}^s = CE_0 \int f_{qp} \exp\{j\Phi\} dx'dy' \tag{16}$$

with the phase term $\Phi = k[(\hat{k}_s - \hat{k}_i)x' + (\hat{k}_s - \hat{k}_i)y' + (\hat{k}_s - \hat{k}_i)\zeta]$, where the third term constitutes a random process because of $\zeta(x, y)$, $C = -jk/4\pi R$.

The Kirchhoff field coefficients $f_{qp}$ appearing in (16) may be more explicitly written in the following form:

$$
\begin{aligned}
f_{vv} = &-[(1 - R_v)\hat{h}_s \cdot (\hat{n} \times \hat{v}) + (1 + R_v)\hat{v}_s \cdot (\hat{n} \times \hat{h})]S_1 \\
&- (R_h + R_v)(\hat{v} \cdot \hat{t})[(\hat{h}_s \cdot \hat{d})(\hat{n} \cdot \hat{k}_i) - (\hat{n} \cdot \hat{d})(\hat{h}_s \cdot \hat{k}_i) - (\hat{v}_s \cdot \hat{t})(\hat{n} \cdot \hat{k}_i)]S_1
\end{aligned} \tag{17}
$$

$$
\begin{aligned}
f_{vh} = &[(1 - R_h)\hat{v}_s \cdot (\hat{n} \times \hat{v}) - (1 + R_h)\hat{h}_s \cdot (\hat{n} \times \hat{h})]S_1 \\
&- (R_h + R_v)(\hat{h} \cdot \hat{d})[(\hat{h}_s \cdot \hat{t})(\hat{n} \cdot \hat{k}_i) - (\hat{n} \cdot \hat{d})(\hat{v}_s \cdot \hat{k}_i) + (\hat{v}_s \cdot \hat{d})(\hat{n} \cdot \hat{k}_i)]S_1
\end{aligned} \tag{18}
$$

$$
\begin{aligned}
f_{hv} = &[(1 - R_v)\hat{v}_s \cdot (\hat{n} \times \hat{v}) - (1 + R_v)\hat{h}_s \cdot (\hat{n} \times \hat{h})]S_1 \\
&- (R_h + R_v)(\hat{v} \cdot \hat{t})[(\hat{h}_s \cdot \hat{t})(\hat{n} \cdot \hat{k}_i) - (\hat{n} \cdot \hat{d})(\hat{v}_s \cdot \hat{k}_i) + (\hat{v}_s \cdot \hat{d})(\hat{n} \cdot \hat{k}_i)]S_1
\end{aligned} \tag{19}
$$

$$
\begin{aligned}
f_{hh} = &[(1 + R_h)\hat{v}_s \cdot (\hat{n} \times \hat{h}) + (1 - R_h)\hat{h}_s \cdot (\hat{n} \times \hat{v})]S_1 \\
&- (R_h + R_v)(\hat{h} \cdot \hat{d})[(\hat{h}_s \cdot \hat{d})(\hat{n} \cdot \hat{k}_i) - (\hat{n} \cdot \hat{d})(\hat{h}_s \cdot \hat{k}_i) - (\hat{v}_s \cdot \hat{t})(\hat{n} \cdot \hat{k}_i)]S_1
\end{aligned} \tag{20}
$$

where the Fresnel reflection coefficients $R_p$ are given in [22,23], $S_1 = \sqrt{1 + \zeta_x^2 + \zeta_y^2}$, $\hat{n}$ is surface normal vector, and $\hat{h}$, $\hat{v}$ and $\hat{h}_s$, $\hat{v}_s$ are the horizontal-polarized and vertical-polarized vector for the incident and scattering waves, respectively.

### 3.2. Scattered Power

Once we have the scattered field estimates, the next step is to compute the scattered power. Noting that the scattered field is composed of a mean-field (coherent) and a fluctuating field (incoherent):

$$E_{qp}^s\left(\vec{r}\right) = E_{qp}^m\left(\vec{r}\right) + E_{qp}^f\left(\vec{r}\right) \tag{21}$$

where $\vec{r}$ is the position vector $\vec{r} = (x, y, z)$, the ensemble average of fluctuating field is zero, and $\left\langle E_{qp}^f\left(\vec{r}\right)\right\rangle = 0$.

The scattered power is given by

$$\left\langle E_{qp}\left(\vec{r_1}\right)E_{qp}^*\left(\vec{r_2}\right)\right\rangle = \left\langle E_{qp}^m\left(\vec{r_1}\right)\right\rangle\left\langle E_{qp}^{m*}\left(\vec{r_2}\right)\right\rangle + \left\langle E_{qp}^f\left(\vec{r_1}\right)E_{qp}^{f*}\left(\vec{r_2}\right)\right\rangle \tag{22}$$

so the coherent term is

$$\left|E_{qp}^m(\vec{r})\right|^2 = \frac{k^2|E_0|^2}{16\pi^2 R^2}|f_{qp}|^2|\langle I\rangle|^2 \tag{23}$$

and the incoherent term is

$$\left|E_{qp}^f(\vec{r})\right|^2 = \frac{k^2|E_0|^2}{16\pi^2 R^2}|f_{qp}|^2\left(\left\langle|I|^2\right\rangle - |\langle I\rangle|^2\right) \tag{24}$$

The integral term appearing in Equations (23) and (24) is

$$I = \iint\limits_{A_0} e^{-i(q_x x + q_y y)} e^{-iq_z \zeta(x,y)} dxdy \tag{25}$$

The ensemble averages we have to evaluate are of the form

$$\langle I\rangle = \iint\limits_{A_0} e^{-i(q_x x + q_y y)}\left\langle e^{-iq_z \zeta}\right\rangle dxdy \tag{26}$$

$$\left\langle |I|^2 \right\rangle = \iint\limits_{A_0} dxdy \iint\limits_{A_0} dx'dy' e^{i(q_x(x-x')-q_y(y-y'))} \left\langle e^{iq_z(\zeta-\zeta')} \right\rangle \tag{27}$$

with wave components

$$\begin{aligned}
q_x &= k(\sin\theta_s\cos\phi_s - \sin\theta\cos\phi) \\
q_y &= k(\sin\theta_s\sin\phi_s - \sin\theta\sin\phi) \\
q_z &= k(\cos\theta_s + \cos\theta)
\end{aligned} \tag{28}$$

To evaluate Equations (26) and (27), we make use of the characteristic functions of the univariate and bivariate distributions, respectively.

The univariate and bivariate characteristics functions, $\phi_u$, $\phi_b$, are defined as:

$$\phi_u \triangleq \left\langle e^{-iq_z z} \right\rangle = \int_{-\infty}^{\infty} e^{-iq_z\zeta} p(\zeta)d\zeta \tag{29}$$

$$\phi_b \triangleq \left\langle e^{-iq_z(\zeta-\zeta')} \right\rangle = \int_{-\infty}^{\infty}\int_{-\infty}^{\infty} d\zeta d\zeta' p(\zeta,\zeta') e^{iq_z(\zeta-\zeta')} \tag{30}$$

Following the approach by Beckmann [17], we obtain the univariate and bivariate characteristics function for a Gaussian height distribution:

$$\phi_{u,g} = e^{-\sigma^2 q_z^2/2} \tag{31}$$

$$\phi_{b,g} = \exp[-q_z^2\sigma^2(1-\rho)] \tag{32}$$

Similarly, the univariate and bivariate characteristics functions for exponential height distribution are

$$\phi_{u,e} = \frac{1}{\sigma}\left(\frac{1}{\sigma} + jq_z\right)^{-1} \tag{33}$$

$$\phi_{b,e} = \frac{1}{1 + q_z^2\sigma^2(1-\rho)} \tag{34}$$

Substituting (31)–(34) into (26) and (27), we compute the coherent and incoherent scattering coefficients for Gaussian and exponential height distributions.

### 3.3. Scattering Coefficients

The scattering coefficients are defined by

$$\sigma_{qp}^0 = \frac{4\pi R^2 P_{qp}}{E_0^2 A_0} \tag{35}$$

where $R$ is the range from surface to observation point, $A_0$ is the effective antenna illuminated area over the surface, $P_{qp}$ is the scattered power, and $E_0$ is the amplitude of the electric field.

After some mathematical manipulations, the coherent and incoherent scattering coefficients are given by

(1) For a Gaussian height distribution surface:

$$\sigma_{qp,coh}^0 = \pi k^2 |f_{qp}|^2 e^{-q_z^2\sigma^2}\delta(q_x)\delta(q_y) \tag{36}$$

$$\sigma_{qp,incoh}^0 = \frac{k^2}{2}|f_{qp}|^2 e^{-\sigma^2 q_z^2}\sum_{n=1}^{\infty}\frac{(\sigma^2 q_z^2)^n}{n!}S^{(n)}(q_x,q_y) \tag{37}$$

(2) For an exponential height distribution surface:

$$\sigma_{qp,coh}^0 = \pi k^2 |f_{qp}|^2 \frac{1}{1 + q_z^2\sigma^2}\delta(q_x)\delta(q_y) \tag{38}$$

$$\sigma^0_{qp,incoh} = \frac{k^2}{2}\left|f_{qp}\right|^2 \frac{1}{1+\sigma^2 q_z^2} \sum_{n=1}^{\infty}\left(\frac{\sigma^2 q_z^2}{1+\sigma^2 q_z^2}\right)^n S^{(n)}\left(q_x, q_y\right) \tag{39}$$

The *n*th-power roughness spectrum is defined as the Fourier transform of the *n*th-power correlation function in Equations (11) and (13):

$$S^{(n)}\left(K_x, K_y\right) = \frac{1}{2\pi}\int_0^{\infty} \rho^n\left(r_x, r_y\right)e^{-j(K_x r_x + K_y r_y)}dr_x dr_y \tag{40}$$

The Gaussian *n*th-power PSD is

$$S_g^{(n)}(\mathbf{K}) = \frac{\ell^2}{2n}\exp\left(-\frac{\mathbf{K}^2 \ell^2}{4n}\right) \tag{41}$$

The exponential *n*th-power PSD is

$$S_e^{(n)}(\mathbf{K}) = \left(\frac{\ell}{n}\right)^2 \left(1+\left(\frac{\mathbf{K}\ell}{n}\right)^2\right)^{-3/2} \tag{42}$$

## 4. Results and Discussion

This section analyzes both the coherent and incoherent bistatic scattering from Gaussian and exponential HPD rough surfaces.

### 4.1. Surface Roughness Dependence

We compared the differences in scattering between Gaussian and exponential HPD rough surfaces. The normalized *RMS* height, $k\sigma$, ranged from 0.1 to 1.2, and the normalized correlation length, $k\ell$, was set to 8, with a permittivity of $15 - j1.5$. In Figure 3a, we note that the coherent scattering from the exponential HPD surface is higher than that from the Gaussian HPD surface and is more pronounced for a larger *RMS* height. The coherent scattering is strongly affected by the surface height distribution. In addition, the HH polarization is higher than that of VV polarization, regardless of the HPDs. The above observation is confirmed with experimental measurements in the visible region [24] that the stronger coherent component exists in the exponential HPD surfaces, even in a rougher (deep phase screen) surface. We note that in [24], only the backscattering at normal incidence was investigated, and no autocorrelation function or PSD was involved.

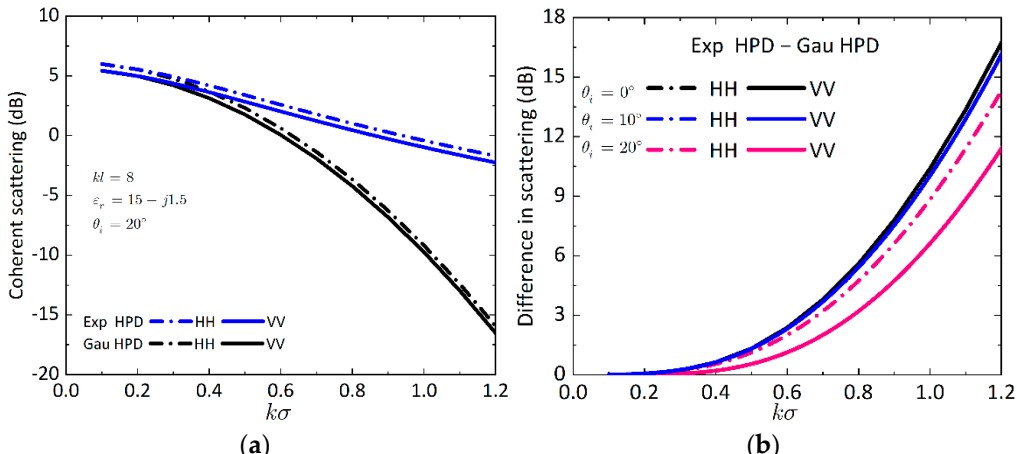

**Figure 3.** (**a**) Comparison of the coherent scattering between Gaussian and exponential HPD rough surface; (**b**) Difference between exponential and Gaussian HPD rough surface.

To better illustrate the non-Gaussian effect on the coherent scattering, we plotted the difference between exponential and Gaussian HPD cases at 0°, 10°, and 20° of incident

angles. In Figure 3b, due to the exponential HPD, the difference in coherent scattering between exponential and Gaussian HPD cases is as wide as about 0~18 dB. The coherent scattering is dramatically enhanced for the exponential HPD surface. As the *RMS* height increases, the differences in coherent scattering grow almost exponentially. The non-Gaussian effect on the coherent scattering is strongest at normal incidence. In addition, a stronger dependence on the surface height distribution effect shows up in HH polarization. These results show that the coherent scattering for both HH and VV polarizations is higher from the exponential HPD surfaces than from the Gaussian HPD surfaces.

Under the same PSD, we evaluated the difference in incoherent scattering between a Gaussian and an exponential HPD surface. Three surface roughness scales were examined for the non-Gaussian effect. In general, the angular trends in the forward region are quite similar for the Gaussian and exponential PSD. However, around the specular direction, as the RMS height increases, the difference of incoherent scattering between Gaussian and exponential HPD surfaces varies in an oscillatory fashion, as shown in Figure 4. Such angular dependence of the roughness was not shown in backscattering [18]. Hence, the backscattering properties cannot generally apply to the whole scattering plane. That is, in the exponential HPD, the incoherent scattering is slightly weakened at a smaller roughness but significantly enhanced at a larger roughness. This phenomenon is even more pronounced for the exponential PSD, as shown in Figure 4b. For the exponential PSD, the difference in incoherent scattering around the specular direction is as large as 10 dB.

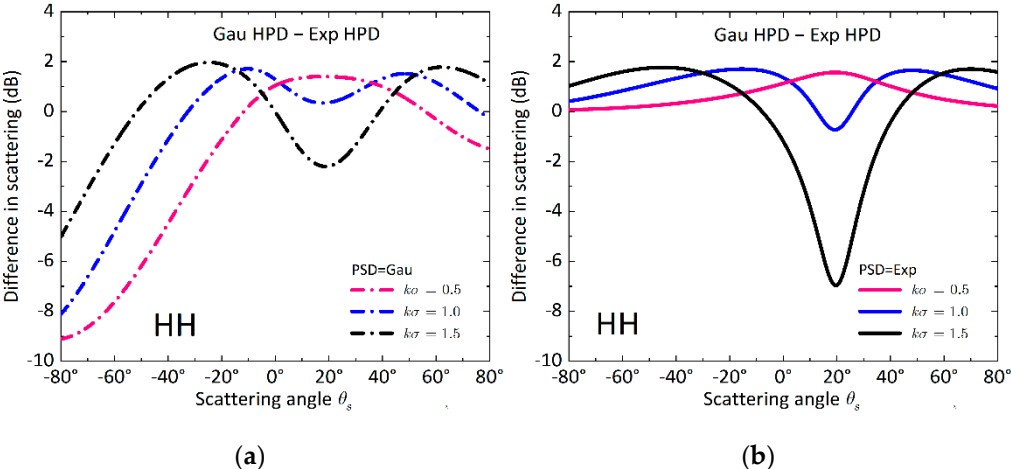

**Figure 4.** Difference of incoherent scattering as function of scattering angle between the Gaussian and exponential HPD rough surfaces with Gaussian and exponential PSDs at $kl = 8$, $\varepsilon_r = 15 - j1.5$, $\theta_i = 20°$, $\phi_s = 0°$. (**a**) Gaussian PSD, (**b**) exponential PSD.

Furthermore, at a larger scattering angle ($\theta_s > 50°$), compared with the Gaussian HPD surface, the incoherent scattering from exponential HPD surfaces is first enhanced and then weakened as the roughness increases. In addition, the angular trends are no longer the same between Gaussian and exponential PSD in the backward region. In the backward region, we note that the difference in incoherent scattering changes from $-9$ dB to about 2 dB for the Gaussian PSD surface but varies from 0 to 2 dB for exponential PSD. The incoherent scattering in the backward region is more sensitive to exponential HPD when the surface is Gaussian PSD. Moreover, the incoherent scattering from the exponential PSD surface is enhanced in the backward region, a fact also reported in [17,18] for backscattering.

To further explore the impact of the surface roughness, we present the difference of incoherent scattering between Gaussian and exponential HPD under three roughness scales in Figure 5. As the scattering azimuthal angle is rotating from the specular direction to the backscattering direction, we can note that the angular trends of the incoherent scattering are quite different under the three roughness scales. The difference in incoherent scattering between the Gaussian and exponential HPD nonlinearly depends on the surface

roughness. Such dependence is greatly affected by the surface PSD. In the forward region, the incoherent scattering is enhanced for smooth surfaces but weakened for rough surfaces. This phenomenon is reversed in the backward region. For a rougher surface, the incoherent scattering in the forward region is more sensitive to exponential HPD.

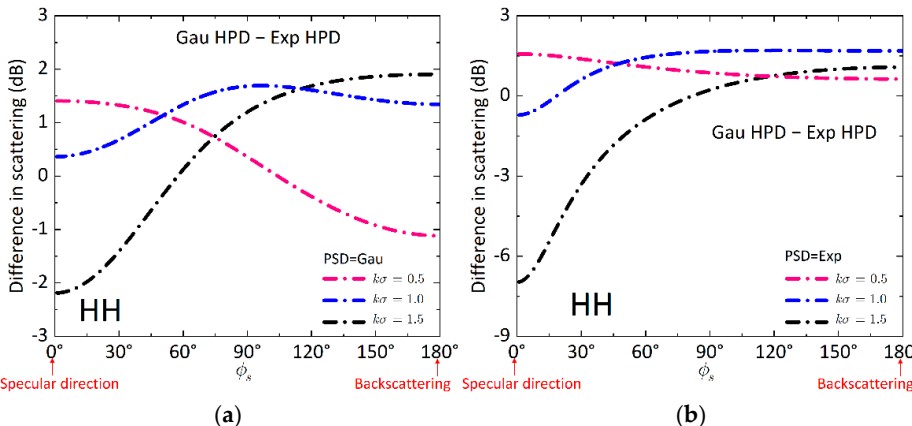

**Figure 5.** Difference of incoherent scattering as function of scattering azimuthal angle between the Gaussian and exponential HPD rough surfaces with Gaussian and exponential PSDs at $kl = 8$, $\varepsilon_r = 15 - j1.5$, $\theta_i = 20°$, $\theta_s = 20°$. (**a**) Gaussian PSD, (**b**) exponential PSD.

### 4.2. Scattering Angular Dependence

Figure 6 shows the incoherent scattering in an incident plane for four surfaces: two HPDs and two PSDs. The incoherent scattering as a function of scattering angle is presented with the surface roughness of $kl = 8$, $k\sigma = 1.0$, and permittivity of $15 - j1.5$. The incident angle was fixed at $20°$. These comparisons show that the non-Gaussian effect on the incoherent scattering is significantly different under Gaussian and exponential PSDs. To explore the effect of surface HPD with exponential PSD, we compared the numerical results for Gau HPD and Exp PSD (blue line) and Exp HPD and Exp PSD (black line). From the incoherent scattering plots of Figure 6, the angular shape and width are more dominated by the PSD than by the HPD. For coherent scattering, the angular width is controlled by the HPD and is wider for a Gaussian HPD surface [24].

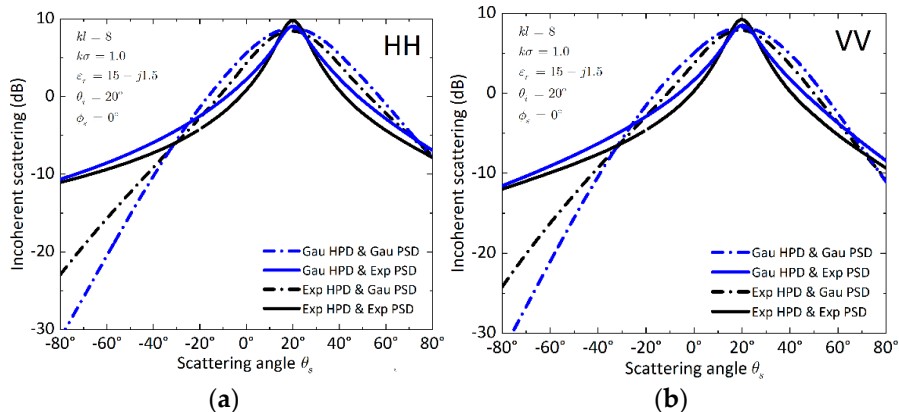

**Figure 6.** Comparison of bistatic incoherent scattering in an incident plane for four types of surfaces: Gau HPD and Gau PSD, Gau HPD and Exp PSD, Exp HPD and Gau PSD, Exp HPD and Exp PSD. The related parameters are $kl = 8$, $k\sigma = 1.0$, $\varepsilon_r = 15 - j1.5$, $\theta_i = 20°$, $\phi_s = 0°$. (**a**) HH polarization, (**b**) VV polarization.

The incoherent scattering from the exponential PSD surface is lower than that from the Gaussian HPD surface, except in the specular direction. Furthermore, for Gaussian PSD, the incoherent scattering is reduced because of exponential HPD in the forward scattering

region ($\theta_s > 0°$). However, in the backward scattering region ($\theta_s < 0°$), the incoherent scattering is weakened at a small scattering angle but enhanced at a larger scattering angle. We continued to compare the incoherent scattering along the azimuthal direction for four surfaces. For numerical illustration, the surface roughness was set to $kl = 8$, $k\sigma = 1.0$. As shown in Figure 7, for the same PSD, the incoherent scattering from the exponential HPD surface is higher than that from the Gaussian HPD surface.

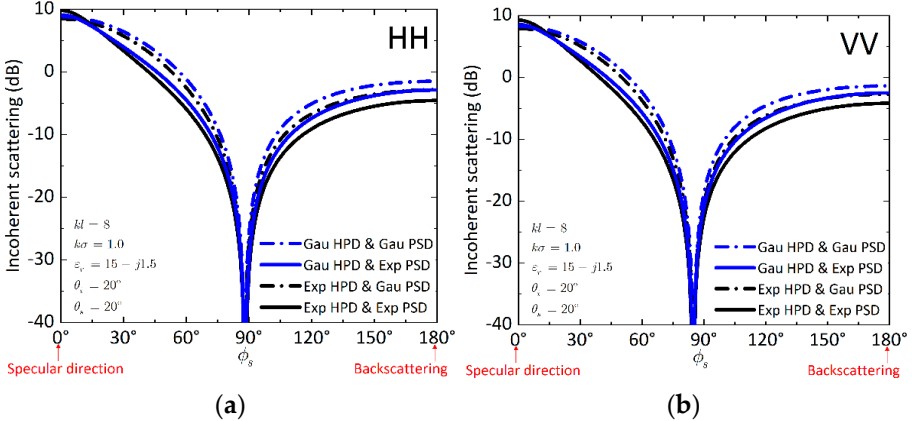

**Figure 7.** Comparison of bistatic incoherent scattering along the azimuthal direction for four types of surfaces: Gau HPD and Gau PSD, Gau HPD and Exp PSD, Exp HPD and Gau PSD, Exp HPD and Exp PSD. The related parameters are $kl = 8$, $k\sigma = 1.0$, $\varepsilon_r = 15 - j1.5$, $\theta_i = 20°$, $\theta_s = 20°$. (**a**) HH polarization, (**b**) VV polarization.

Next, we compared the hemispherical plots of bistatic scattering on the whole scattering plane between Gaussian and exponential HPD rough surfaces. As shown in Figure 8, the left-half and right-half regions of the hemispherical plots correspond to the backward and forward regions, respectively. To examine the non-Gaussian effect, we plotted the bistatic scattering by fixing the PSD as Gaussian. The surface parameters are $kl = 8$, $k\sigma = 1.0$, $\varepsilon_r = 15 - j1.5$, with the incident angle of $20°$. As a reference, the scattering patterns for Gaussian HPD with Gaussian PSD are given in Figure 8a.

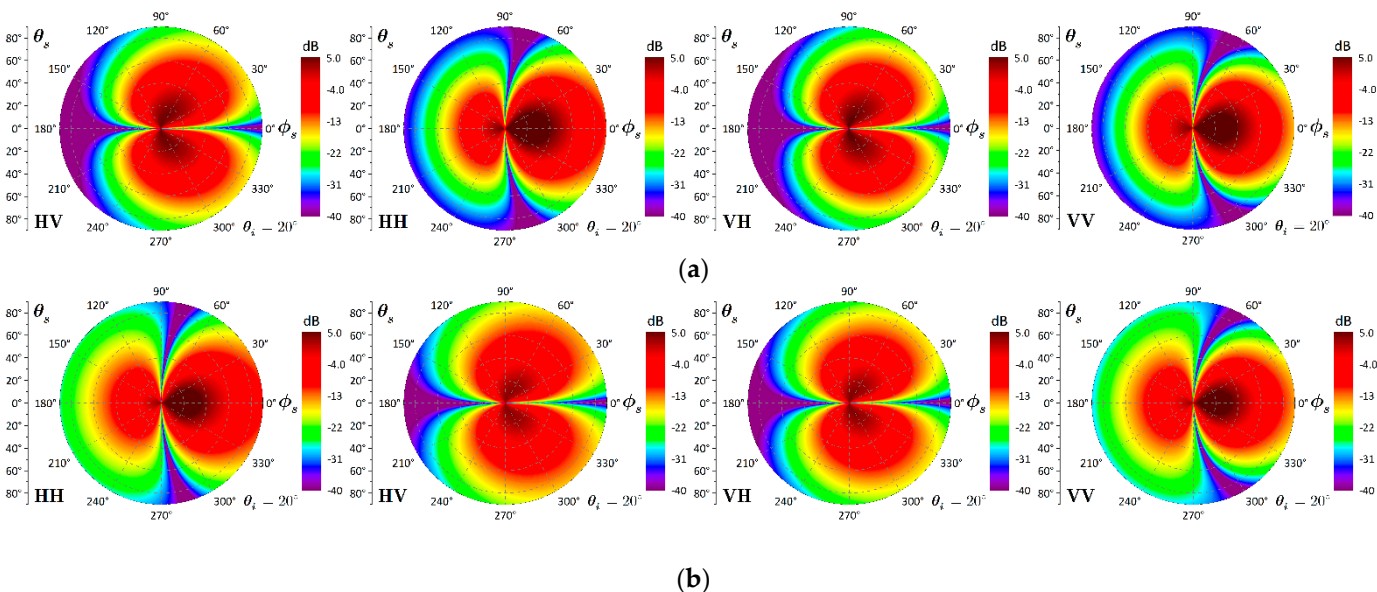

**Figure 8.** Bistatic scattering from the Gaussian and exponential HPDs with Gaussian PSD: $kl = 8$, $k\sigma = 1.0$, $\varepsilon_r = 15 - j1.5$, $\theta_i = 20$. (**a**) Gaussian HPD and Gaussian PSD (Gau HPD and Gau PSD), (**b**) exponential HPD and Gaussian PSD (Exp HPD and Gau PSD).

The co- and cross-polarized scattering coefficients are significantly enhanced in the backward region but slightly weakened in the forward region when comparing Figure 8a,b. As mentioned before, the effect of surface height distribution varies with different PSD. To demonstrate this effect, we set the PSD to exponential and plotted the bistatic scattering patterns from a Gaussian and exponential HPD surface in Figure 9. In virtue of exponential HPD, we can note that the co- and cross-polarized scattering is enhanced on the whole scattering plane except for the specular direction and its vicinity. These results suggest (under the exponential PSD and without considering the exponential HPD) that the bistatic scattering will be overestimated except in the specular direction.

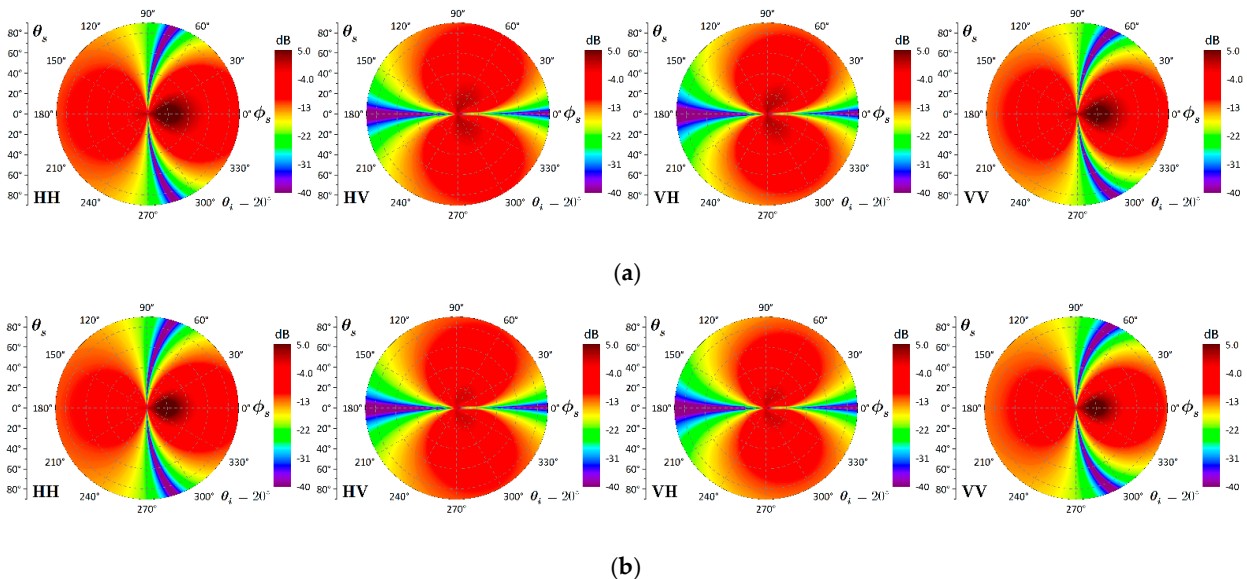

**Figure 9.** Bistatic scattering from the Gaussian and exponential HPDs with the exponential PSD: $kl = 8, k\sigma = 1.0, \varepsilon_r = 15 - j1.5, \theta_i = 20$. (**a**) Gaussian HPD and exponential PSD (Gau HPD and Exp PSD), (**b**) exponential HPD and exponential PSD (Exp HPD and Exp PSD).

The above observation exhibits the effect of surface height distribution on bistatic scattering by fixing the roughness scale. Here, we investigated these effects under three roughness scales. As shown in Figures 10–12, the scattering patterns display the difference, in dB, between Gaussian and exponential HPD cases. In Figure 10, we first fix the PSD as Gaussian. By contrast, the scattering in the backward region is more sensitive to the HPD. For a relatively small roughness ($k\sigma = 0.5$), in virtue of the non-Gaussian HPD, the scattering is enhanced except for the specular direction and its vicinity.

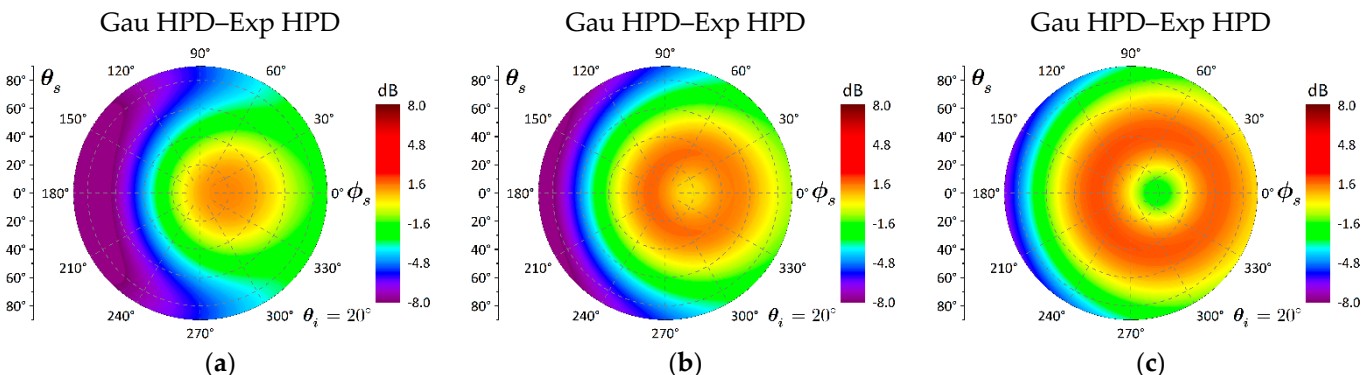

**Figure 10.** Difference of bistatic scattering between Gaussian and exponential HPD rough surface with Gaussian PSD $kl = 8, k\sigma = 0.5, 1.0, 1.5, \varepsilon_r = 15 - j1.5, \theta_i = 20$. (**a**) $k\sigma = 0.5$, (**b**) $k\sigma = 1.0$, (**c**) $k\sigma = 1.5$.

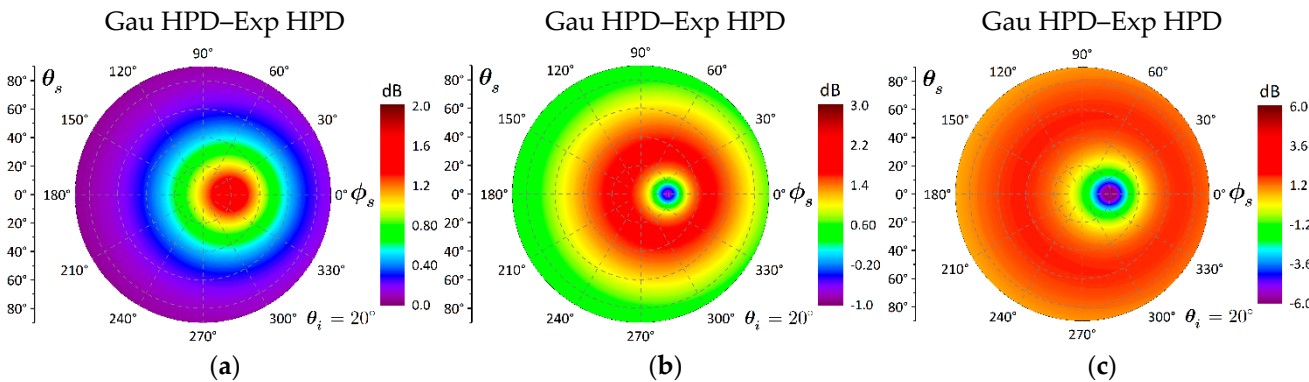

**Figure 11.** Difference of bistatic scattering between Gaussian and exponential HPD rough surface with exponential PSD $kl = 8, k\sigma = 0.5, 1.0, 1.5, \varepsilon_r = 15 - j1.5, \theta_i = 20$. (**a**) $k\sigma = 0.5$, (**b**) $k\sigma = 1.0$, (**c**) $k\sigma = 1.5$.

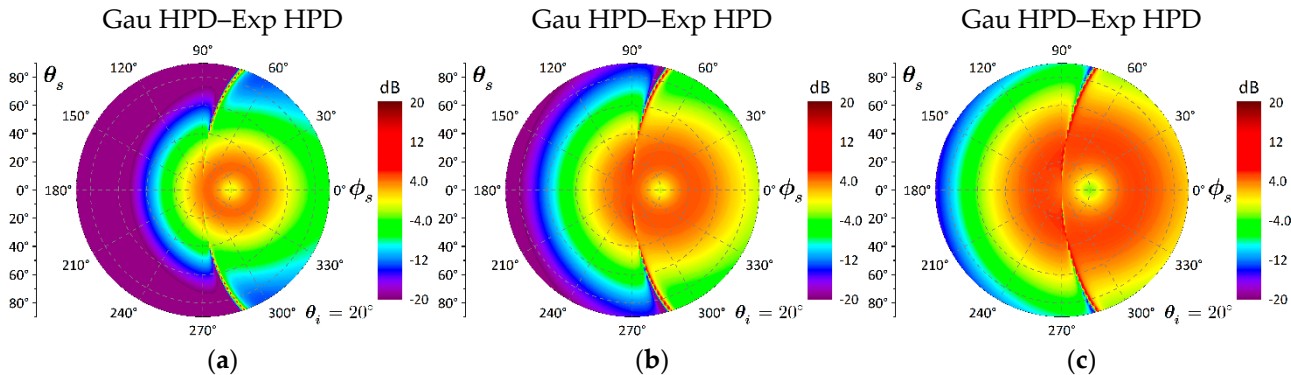

**Figure 12.** Difference of bistatic scattering between Gaussian HPD surface with Gaussian PSD and exponential HPD with exponential PSD $kl = 8, k\sigma = 0.5, 1.0, 1.5, \varepsilon_r = 15 - j1.5, \theta_i = 20$. (**a**) $k\sigma = 0.5$, (**b**) $k\sigma = 1.0$, (**c**) $k\sigma = 1.5$.

Moreover, the backward scattering for the Gaussian HPD surface is about 8 dB less than that for the exponential HPD surface. As the surface roughness increases, we may summarize the variation of the scattering patterns as follows. Due to the non-Gaussian HPD, the scattering in the forward scattering region is reduced except in the specular direction, implying that the coherent scattering is enhanced, especially from a rougher surface. In the backward scattering region, the non-Gaussian HPD enhances the scattering coefficient at a larger scattering angle ($\theta_s > 45°$) but decreases the scattering coefficient at a smaller scattering angle ($\theta_s \leq 45°$). In addition, the difference of backward scattering between Gaussian and exponential HPD cases decreases, but that difference increases in the forward region. That is to say, the sensitivity of backward scattering to HPD is diminished, but that of forward scattering to HPD is enhanced.

As another example, we plotted the difference of bistatic scattering between Gaussian and exponential HPD rough surface with exponential PSD, as shown in Figure 11. The relevant parameters are the same as those in Figure 10. By comparison, when the PSD is switched from Gaussian to exponential, the effect of surface height distribution on bistatic scattering is quite different from Figure 10. In the case of exponential HPD, when the roughness is relatively small, the scattering is reduced on the whole scattering plane. As the surface roughness increases, the exponential HPD weakens the scattering in almost all directions but enhances the scattering in the specular direction and its vicinity. The dynamic range of bistatic scattering is about −6 dB~6 dB in virtue of exponential HPD. That is, the effect of surface height distribution on bistatic scattering is enhanced as the surface roughness increases.

We then examined the coupling effect of HPD and PSD on bistatic scattering in Figure 12. It is important to note that the dynamic range of bistatic scattering was −20~20 dB when both the HPD and PSD were set to the exponential—noting that the effects of HPD and PSD strongly depend on the surface roughness. In the virtual of the non-Gaussian effect, for the relatively small roughness, the notable increase is concentrated in the forward region. However, the apparent decrease is located in the backward region. As the surface roughness increases, the bistatic scattering increases on the whole scattering plane. Moreover, the difference between the scattering coefficients in the forward region and backward region is relatively weaker. This phenomenon indicates that when the surface roughness is larger, the coupling effect of HPD and PSD between forward scattering and backward scattering is reduced.

## 5. Conclusions

Bistatic scattering from Gaussian and exponential height distributed rough surfaces was investigated by the Kirchhoff theory, which only accounts for single scattering. Numerical results show that the non-Gaussian effect on coherent and incoherent scattering is quite different. The coherent scattering from the non-Gaussian HPD surface is higher than that from the Gaussian HPD surface. As the normalized RMS height varies from 0 to about 1.2, the dynamic range of differences between non-Gaussian and Gaussian HPD surfaces is about 0~18 dB. That is, the non-Gaussian height distribution enhances the coherent scattering. As the RMS height increases, their differences in coherent scattering increase almost exponentially. Moreover, the non-Gaussian effect on the coherent scattering is highest at normal incidence and decreases as the incident angle increases. By contrast, the non-Gaussian height effect is significant in HH polarization. The results also confirm that the non-Gaussian effect on incoherent scattering is distinct for Gaussian and exponential PSDs.

The non-Gaussian effect is exhibited by fixing the PSD. Due to the non-Gaussian HPD, the variation of the incoherent scattering characteristic can be summarized in the following points. For the Gaussian PSD surface, the incoherent scattering coefficients are significantly enhanced in the backward region but slightly weakened in the forward region. As the surface roughness increases, the forward scattering is reduced except in the specular direction. The backward scattering increases at a larger scattering angle but decreases at a smaller scattering angle. The difference between forward and backward scattering is shrunk, especially at more significant roughness.

However, for the non-Gaussian PSD surface, the incoherent scattering is enhanced on the whole scattering plane except for the specular direction and its vicinity. When the surface becomes rougher, the scattering weakens in almost the whole scattering plane for the non-Gaussian HPD; however, it enhances the forward scattering in the specular direction and vicinity. The results suggest that the effect of non-Gaussian height on bistatic scattering is enhanced as the surface roughness increases. Furthermore, the difference between coherent and incoherent scattering is widened because of the non-Gaussian height effect.

This paper analyzed the scattering behavior from Gaussian and exponential height distributions. It would be worth examining the degree of non-Gaussianity's impact on the coherent and incoherent scattering and their ratio, and hence, in some essence, the fading strength. Another subject for future study is the examination of the circularly polarized bistatic scattering from a non-Gaussian height distributed surface and its application to GNSS–R.

**Author Contributions:** Conceptualization, K.-S.C.; methodology, K.-S.C. and Y.Y.; software, Y.Y. and S.W.; validation, K.-S.C. and Y.Y.; formal analysis, Y.Y. and S.W.; investigation, K.-S.C., Y.Y. and S.W.; resources, Y.Y.; data curation, Y.Y. and S.W.; writing—original draft preparation, K.-S.C. and Y.Y.; writing—review and editing, K.-S.C. and Y.Y.; visualization, Y.Y. and S.W.; supervision, K.-S.C.; project administration, K.-S.C.; funding acquisition, K.-S.C. All authors have read and agreed to the published version of the manuscript.

**Funding:** This work was supported by the Fundamental Research Funds for the Central Universities No. 30922010311, Guangxi Natural Science Fund for Innovation Research Team under Grant 2019GXNSFGA245001, and Guangxi Natural Science Youth Fund under Grant 2020GXNSFBA297105.

**Conflicts of Interest:** The authors declare no conflict of interest.

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
