# Peer review of "Bistatic Radar Scattering from Non-Gaussian Height Distributed Rough Surfaces"

_remotesensing, doi:10.3390/rs14184457_

Round 1

Reviewer 1 Report

This paper use the Kirchhoff approximation to study the first order bistatic scattering of non-Gaussian surfaces. A difference as large as 18dB is observed between Gaussian and Exponential distributed heights. Non-Gaussian effects are significant for surfaces with exponential correlation functions.

Here are some questions and suggestions I have:

1.       In the paper, surfaces with different PSD and height distributions are studies. Could be added to explain the concept in more details? For example, some figures of the 1D surface profiles and PSDs for different combinations.

2.       It would be useful to explain in more details for the non-Gaussian surfaces and compare to the conventional Gaussian surface.

3.       Regarding the method, only Kirchhoff model is used. Is the model valid for the study?

4.       Figure 4 and 5 should be moved to later part of the paper, it is better to present the incoherent bistatic scattering results first.

Reviewer 2 Report

Theoretical modeling of surface scattering problems is the foundation of microwave remote sensing. The authors did an interesting job analyzing the impacts of different surface roughness patterns on bi-static scattering. I like the work, but also suggest the authors address the following issues before the publication:

(a) I believe numerical or experimental validations of the KA-based study are needed for strengthening the reliability of the roughness impact analysis.

(b) Will it be possible to provide an improved model version for not only calculating scattering intensity but also wave phases? This will be useful for GNSS-IR applications.

(c) Maybe I missed something for Figures 8 and 9, but I assume no cross-polarized signals are supposed to show up in these single-scattering simulations.

(d) Could the authors also discuss the bi-static scattering for non-linear polarizations (e.g. circular polarization commonly used for GNSS-R applications)?

Reviewer 3 Report

This paper analyses the scattering from a rough surface. Two distribution functions were used for the surface height: the Gaussian function and the exponential function. The same functions were used for the power spectrum density. Angular dependences are contrasted. Moreover, height probability densities and power spectrum densities are combined to present the impact of such combinations on the scattering obtained. Consequently, this is a theoretical analysis that could be published in Remote Sensing after the introduction of some major changes.

A key problem of the paper lies on its structure. A scientific paper is formed by the abstract, keywords, introduction, materials and methods, results, discussion, and conclusions. This analysis presents results, although with another name and it does not present discussion. In the discussion section, which could be combined with the results section, the authors should compare their results with those of other studies. Perhaps new references should be included or some of the introduction references could be removed and employed for discussion.

The authors have focused on a couple of distribution functions. They indicated that the Gaussian one is usual. However, they should indicate the reasons to select the exponential function against other distributions, or even if other distributions could be employed.

Since the surface features are quite varied, the authors should provide some information about the surfaces suitable for each distribution function. Moreover, this is a theoretical analysis that should be contrasted with experimental data if this is possible.

Finally, a paragraph with future research lines would be recommended at the end of the conclusions.

Minor remarks.

All the acronyms must be introduced.

Line 89. Replace “Distrubution” by “Distribution”.

Round 2

Reviewer 2 Report

Thanks for the clarifications. I look forward to the further development of the  bi-static scattering modeling . 

Reviewer 3 Report

The reviewer’s comments were considered. However, all the acronyms should be revised since they should be introduced the first time that they were cited.